# Biochemical Identification and Clinical Description of Medetomidine Exposure in People Who Use Fentanyl in Philadelphia, PA

**DOI:** 10.3390/ijms26146715

**Published:** 2025-07-13

**Authors:** Phil Durney, Jennifer L. Kahoud, TaReva Warrick-Stone, Maeve Montesi, Meg Carter, Sabrina Butt, Alberto Martinez Mencia, Louisa Omoregie, Monali Shah, Mariah Bloomfield, Nicholas Tomasko, Rebecca Jaffe, Allison Herens, Warren R. Korn, Karen Alexander, Douglas Stickle, Dennis Goodstein, Lara Carson Weinstein, Kory S. London

**Affiliations:** 1Department of Internal Medicine, Thomas Jefferson University, 1020 Walnut Street, Philadelphia, PA 19107, USA; philip.durney@jefferson.edu (P.D.); margaret.carter@jefferson.edu (M.C.); rebecca.jaffe@jefferson.edu (R.J.); 2Jefferson Addiction Multidisciplinary Service, Thomas Jefferson University, 1020 Walnut Street, Philadelphia, PA 19107, USA; tareva.warrick-stone@jefferson.edu (T.W.-S.); allison.herens@jefferson.edu (A.H.); dennis.goodstein@jefferson.edu (D.G.); lara.weinstein@jefferson.edu (L.C.W.); 3Sidney Kimmel Medical College, Thomas Jefferson University, 1020 Walnut Street, Philadelphia, PA 19107, USA; jennifer.kahoud@jefferson.edu (J.L.K.); mariah.bloomfield@students.jefferson.edu (M.B.); nicholas.tomasko@students.jefferson.edu (N.T.); 4Department of Emergency Medicine, Thomas Jefferson University, 1020 Walnut Street, Philadelphia, PA 19107, USA; sabrina.butt@jefferson.edu (S.B.); louisa.omoregie@jefferson.edu (L.O.); monali.shah@jefferson.edu (M.S.); 5Department of Pathology, Thomas Jefferson University, 1020 Walnut Street, Philadelphia, PA 19107, USA; alberto.martinezmencia@jefferson.edu (A.M.M.); warren.korn@jefferson.edu (W.R.K.); douglas.stickle@jefferson.edu (D.S.); 6Department of Family and Community Medicine, Thomas Jefferson University, 1020 Walnut Street, Philadelphia, PA 19107, USA; 7Friends Research Institute, 1516 N. 5th St., Suite 321, Philadelphia, PA 19122, USA; kalexander@friendsresearch.org

**Keywords:** medetomidine, fentanyl, adulterant, LC-MS/MS, metabolites, glucuronidase, opioid withdrawal, xylazine, substance use disorder, human metabolism

## Abstract

Medetomidine, a veterinary α2-adrenergic agonist, has recently emerged as an adulterant in the non-medical opioid supply, yet human exposure has remained poorly characterized. We conducted a pragmatic retrospective cohort analysis utilizing chart review and liquid chromatography–tandem mass spectrometry (LC-MS/MS) toxicology testing on available urine samples from patients presenting to two hospitals in Philadelphia, PA, who fit two clinical phenotypes, intoxication or withdrawal. Samples also underwent glucuronidase pre-treatment to assess impact on the yield of medetomidine and xylazine metabolite detection. Testing identified universal exposure to medetomidine (58/58 samples) via the 3-hydroxy-medetomidine (3-OH-M) metabolite, post glucuronidase treatment and variable xylazine exposure (40/58 samples). Importantly, 32% of medetomidine exposures would have been missed without enzymatic pre-treatment. Patients exhibited two distinct clinical phenotypes: intoxication, characterized primarily by sedation; bradycardia; and often hypotension, and withdrawal, presenting with life-threatening tachycardia; hypertension and often encephalopathy. Notably, clinical phenotype correlated with urinary concentrations of 3-OH-M but not xylazine. These findings underscore the critical need for heightened clinical awareness and need for contemporaneous toxicologic screening mechanisms for medetomidine exposure, emphasizing its distinct clinical presentations and the potential public health implications posed by its widespread adulteration in illicit opioids.

## 1. Introduction

The ongoing adulteration of pharmaceutical products in the illegal drug supply poses significant public health and safety risks in many communities. Recently, the incorporation of medetomidine—a potent, nonopioid sedative—transformed the fentanyl supply in Philadelphia, leading to severe medical complications from a novel withdrawal syndrome [1]. Drug checking programs and forensic laboratories first detected medetomidine in street opioid samples in 2022 [2]. By 2023–2024, medetomidine was identified in multiple U.S. states and Canada as a co-contaminant in fentanyl or heroin preparations [3]. One of Pennsylvania’s drug checking programs, PAGroundhogs (PAG), confirmed that 61% of the total dope samples tested since April 2024 (169 samples) showed positivity for medetomidine and had surpassed xylazine, which was the previous most prevalent adulterant (present in up to 99% of fentanyl samples in 2023) [4]. Many medetomidine-positive samples in Philadelphia also contained xylazine and fentanyl, indicating that medetomidine is entering the street supply as part of the evolving mixture often referred to as tranq (traditionally associated with xylazine–fentanyl combinations) and now referred to as demon (fentanyl–medetomidine ± xylazine combination) [5]. Similar findings were reported in alerts and media reports of overdose clusters outside of Philadelphia, spreading west in 2024, prompting public health warnings about an “emergent adulterant” driving a new wave of overdoses [3].

Managing novel adulterant exposure and associated medical sequelae has been increasingly difficult for clinicians around the world. This challenge is twofold: first, intoxication and withdrawal syndromes are now far more severe and complex; second, it is increasingly difficult to identify which adulterants are causing specific clinical effects. Both of these challenges have been amplified by the introduction of medetomidine. Capturing and defining the temporal relationship of medetomidine intoxication to withdrawal has been much more challenging than other traditional substances due to its rapid metabolism in humans [6]. In the intoxication phase, patients present with profound sedation with slow heart rate and low blood pressure, yet with relatively preserved respiratory drive (absent co-ingestions). This phase quickly escalates with patients exhibiting a severe combination of opioid and sedative withdrawal symptoms. Similar symptoms have been documented in critical care case reports describing a well-documented dexmedetomidine withdrawal syndrome featuring agitation, tachycardia, and hypertension when prolonged infusions are stopped abruptly [7,8,9]. Such clinical observations highlight the need to better characterize medetomidine’s acute and post-acute effects in humans.

This study addresses the emerging problem by investigating a series of patients in Philadelphia, PA, who were found to have medetomidine exposure in the context of non-medical fentanyl use. We present a retrospective cohort of 101 emergency department (ED) and hospitalized patients (September 2024–April 2025) who demonstrated atypical phenotypes of intoxication or withdrawal syndromes, prompting liquid chromatography tandem–mass spectrometry (LC-MS/MS) evaluation for novel adulterants in 58 urine samples. Our hypothesis was that patients presenting with intoxication phenotypes would have high concentrations of medetomidine and fentanyl metabolites with variable xylazine exposure, and withdrawal states would have lower medetomidine and fentanyl metabolite concentrations with, again, variable xylazine exposure. We present the results of clinical and urine toxicology variables collected during intoxication and withdrawal. The objective of this study is to describe the distinct phenotypes associated with medetomidine exposure (in combination with fentanyl and other substances) and describe the clinical findings and toxicology studies of patients with these presentations.

## 2. Results

### 2.1. Study Cohort Description

A total of 101 patients met criteria for inclusion during the six-month study period from September 2024 to April 2025. The median age of patients was 39 years (IQR: 35–44.5), and 74% were male. All patients had a documented history of non-medical fentanyl use and/or tested positive for norfentanyl through standard urine drug screening. The cohort was identified based on clinical suspicion for intoxication or withdrawal phenotype that prompted specialized toxicologic analysis for novel adulterants.

The majority of cases (88 patients, 85.4%) presented with a withdrawal phenotype. Thirteen patients (12.6%) presented with an intoxication phenotype. Within this subgroup, five patients had delayed urine collection, potentially impacting detection sensitivity, and were analyzed separately along with the withdrawal cohort (as all were withdrawing during urine acquisition). Additionally, two patients provided ‘Paired’ urine samples: one obtained during distinct intoxication and another during withdrawal states during the same hospital stay, permitting direct pharmacologic comparison between clinical states. These are shown separately. Table 1 below provides an overview of key characteristics of the intoxication versus withdrawal presentation groups in the cohort:

### 2.2. Toxicology and Analytical Findings

Standard urine immunoassay toxicology was eventually performed in 100/101 patients (99%). See Table 2 for full standard immunoassay results. Screening demonstrated widespread multi-substance exposure across the cohort. Fentanyl was detected in 100% of patients with available results, consistent with the primary inclusion criteria. Cocaine was identified in 60% of individuals, with other opiates and benzodiazepines also commonly present in 40% and 37% of tested individuals, respectively, and nearly a quarter of individuals were positive for both methadone and fentanyl.

LC-MS/MS toxicology results were available for fifty-eight samples: fifty in the withdrawal group and eight in the intoxication group. Forty-three of the patients in the withdrawal group presented with a primary withdrawal phenotype, five came from the primary intoxication phenotype, but whose urine was not collected until the patient had already transitioned into a withdrawal phenotype, and two had paired samples of both intoxication and withdrawal. Binary detection was defined as analyte concentrations greater than or equal to 1 ng/mL. See Table 3 for full toxicology data with comparisons between withdrawal and intoxication phenotypes.

Unmetabolized fentanyl was detected in 100% of intoxication cases and 68% of withdrawal cases (*p* = 0.092). Median concentrations were significantly higher in intoxicated patients compared to those in withdrawal (1107 ng/mL vs. 39 ng/mL; *p* = 0.001). Norfentanyl was detected in 100% of both groups, with significantly elevated concentrations in intoxication samples (9666 ng/mL) compared to withdrawal (741 ng/mL; *p* = 0.001).

The medetomidine metabolite 3-OH-M was detected in 100% of intoxicated samples and 68% of withdrawal samples (*p* = 0.092). Median concentrations were markedly higher in intoxication (561 ng/mL) versus withdrawal (13 ng/mL; *p* < 0.001). When glucuronidase pretreatment was performed, 3-OH-M was found in all intoxicated and withdrawal patients (100% in both groups), with significantly higher concentrations among intoxicated patients (8423 ng/mL vs. 34 ng/mL; *p* < 0.001).

Xylazine was present in 50% of intoxicated and withdrawal cases prior to enzymatic treatment (*p* = 1.000). Median concentrations did not differ significantly between groups’ phenotypes (4 ng/mL vs. 3 ng/mL; *p* = 1.000). Glucuronidase pre-treatment resulted in xylazine identification in 100% of intoxication cases, and 64% of withdrawal cases (*p* = 0.48), though concentrations were similar between phenotypes (*p* = 0.272). Table 3 displays the binary identification outcomes, and quantitative LC-MS/MS concentration toxicology results, using median values given the non-normative distribution of the data.

Among the 42 samples positive for both 3-OH-M and its post-glucuronidase metabolite (≥1 ng/mL), there was a significant increase in concentration after glucuronidase treatment. Median 3-OH-M concentration before glucuronidase treatment was 79 ng/mL (IQR: 20–193 ng/mL), which increased significantly to a median of 151 ng/mL (IQR: 42–476 ng/mL) post-glucuronidase (*p* < 0.001). Importantly, while all patients with intoxication screened positive without glucuronidase testing, sixteen (32%) patients in withdrawal tested negative without pretreatment, indicating a large cohort whose exposure may be missed without enzymatic preparation.

Among the 58 specimens tested for xylazine, 29 (50%) were positive (≥1 ng/mL) prior to glucuronidase hydrolysis. Following enzymatic treatment, 40 specimens (69%) were positive. Among the 29 paired specimens with quantifiable concentrations both pre- and post-glucuronidase, the median xylazine concentration increased from 36.0 ng/mL (IQR: 8.0–146.0 ng/mL) to 73.0 ng/mL (IQR: 11.0–145.0 ng/mL) after enzymatic treatment. This difference, however, was not statistically significant (*p* = 0.367).

Lastly, the paired cohort (*n* = 2) who provided two samples each were analyzed for the temporal relationship between their intoxication and withdrawal phenotype samples. The limited sample size and uncontrolled substance use patterns make formal metabolic evaluation infeasible, but it is important to note the rapid elimination of medetomidine, fentanyl, and norfentanyl. See Table 4 for full analyte data.

### 2.3. Clinical Outcomes

Among all 101 patients in the study cohort, treatment of α2-agonist withdrawal was common and intensive, occurring in 100 patients (one patient with intoxication was discharged prior to withdrawal occurrence). See Table 5 for an account of clinical treatment and outcomes. Oral clonidine was administered in 86.4% of cases, and a transdermal clonidine patch was used in 75.7%. Dexmedetomidine infusion was employed in 61.2% of patients.

Cardiovascular and neurologic complications were notable. Elevated troponin levels consistent with non-ST elevation myocardial infarction (NSTEMI) occurred in 20% of patients. A diagnosis of encephalopathy (suspected in some cases to represent posterior reversible encephalopathy syndrome (PRES), due to uncontrolled hypertension in the setting of a non-hypertensive baseline) was documented in 31% of the cohort.

The severity of opioid withdrawal was high. The median Clinical Opiate Withdrawal Scale (COWS) score was 23.0 (IQR: 19.0–28.0), reflecting an average of moderately severe withdrawal symptoms in this population.

## 3. Discussion

This study is one of the first to characterize the toxicology profiles and phenotypic syndromes of patients exposed to medetomidine in the context of non-medical fentanyl use. Through detailed chart review and mass spectrometry analysis, we document distinct toxidrome associated with medetomidine, including profound sedation, bradycardia, and hypotension in intoxicated individuals, as well as rapid shift to hypertensive crisis and refractory withdrawal. Described in detail below, these clinical features diverge meaningfully from those observed with opioid, benzodiazepine, or xylazine exposures and carry important implications for emergency care, toxicology detection, and withdrawal management.

### 3.1. Detecting the Shift from Xylazine to Medetomidine

Xylazine exposure has allowed clinicians to become familiar with α2-agonist withdrawal, characterized by rebound sympathetic hyperactivity and clinical symptoms such as anxiety, restlessness, and insomnia. [10,11]. Initial treatment guidelines for xylazine centered on stabilizing the α2 receptor with agents like tizanidine, guanfacine, and clonidine, providing relief from withdrawal [12,13,14,15]. In 2024, even with successful approaches to managing α2-withdrawal, a new clinical trend emerged: patients began experiencing a rapid and severe withdrawal syndrome, often not responding to conventional oral α2 agents at typical doses [1,16]. This led to speculation that a stronger adulterant entered the drug supply. Subsequent investigation confirmed that medetomidine has a tangible and significant clinical impact on both people who use opioids and the clinical teams that take care of them. From a public health perspective, the rise in medetomidine appears to follow the same trajectory as xylazine, first emerging in the Northeast US and then disseminating to other regions [3,17]. In the future, medetomidine could replace xylazine in some markets, in others, both may disappear. Therefore, continuous surveillance and an understanding of medetomidine’s effects are critical.

### 3.2. Toxicology Testing

Current standard urine drug screens are not designed to detect medetomidine or its metabolites, highlighting a critical gap in clinical toxicology. This reinforces the utility of LC-MS/MS testing in uncovering underrecognized contributors to polysubstance exposure and withdrawal. Importantly, we identified medetomidine’s major metabolite, 3-hydroxy-medetomidine, in the majority of samples, but glucuronidase pre-treatment allowed universal detection of medetomidine exposure and also increased the yield of xylazine exposure screening. This suggests that these analytes and enzymatic pretreatment may serve as future targets for a confirmatory study. Importantly, glucuronidase pre-treatment increased the yield of testing for both medetomidine and xylazine, especially in the cohort suffering from withdrawal. It is possible that direct LC-MS/MS assessment of the glucuronide metabolites would have provided the same results, but this testing was not in the scope of this study. Future studies should assess the viability and cost of glucuronidase pre-treatment vs. direct assessment of glucuronide-conjugated metabolites. Given the rapid onset of withdrawal in some cases, this carries significant concern for future forensic testing protocols.

Understanding the metabolism of medetomidine is important for interpreting toxicology results and anticipating the duration of its clinical effects. Since medetomidine is not approved for humans, the available information is inferred from animal studies and by analogy to dexmedetomidine. Both medetomidine and dexmedetomidine share the same structure except for chirality, so their metabolism is similar [18,19,20,21,22,23,24,25]. Medetomidine is primarily metabolized in the liver by cytochrome P450 enzymes and by conjugation pathways. A key enzyme, CYP2A6, has been shown to mediate aliphatic hydroxylation of medetomidine, where the hydroxylation occurs at the 3-position (on the N-methyl group attached to the imidazole ring) as a dominant metabolic route, yielding 3-OH-M [20]. In our cohort, the detection of 3-OH-M in urine corroborates that this pathway is active in humans as well.

Medetomidine metabolites also undergo phase II metabolism via glucuronidation. Dexmedetomidine is known to be directly N-glucuronidated at the imidazole ring nitrogen by UGT enzymes (notably UGT2B10 and UGT1A4) [21,22]. This produces at least two distinct glucuronide isomers (N3-glucuronide and N1-glucuronide dexmedetomidine). Medetomidine likely forms similar glucuronides. Additionally, after 3-hydroxylation, there is potential for O-glucuronidation of the 3-hydroxy metabolite. The final result is that little medetomidine is excreted unchanged (less than 5%)—instead, the urine contains mainly metabolites. This is consistent with our finding that parent medetomidine was negligibly detectable in urine (and not included in the final analysis), whereas a hydroxylated metabolite was found in all cases receiving LC-MS/MS testing (and β-glucuronidase pre-treatment). Further research is needed to confirm metabolic transformation and investigate the temporal analysis of medetomidine exposure in humans.

The rapid metabolism of medetomidine explains its relatively short duration of action despite high potency. In veterinary use, medetomidine’s sedative effect lasts hours, and in humans, dexmedetomidine infusions reach steady state quickly and are cleared within a few hours after cessation. Our clinical observations of sedation waning in several hours and withdrawal emerging within hours of last use support the thought that the drug does not have a long terminal half-life.

From a toxicology testing standpoint, our results continue to emphasize that targeting metabolites is essential. Most hospital labs currently do not have the availability of LC/MS-MS testing, let alone testing that provides contemporaneous results. Given the increasing prevalence, there is therefore a convincing argument for developing immunoassay tests for commercial use. The difficulty is noted in the literature: immunoassay test strips for xylazine have been developed, but none exist yet for medetomidine [26]. Until testing becomes more accessible, diagnosis of medetomidine exposure will rely on clinical recognition and perhaps LC-MS/MS send-out testing (as we utilized). Our approach of enzymatic pre-treatment and LC-MS/MS metabolite testing allowed us to detect exposure; labs aiming to detect medetomidine exposure via immunoassay might similarly focus on its glucuronidated and hydroxylated metabolites for greater sensitivity.

### 3.3. Intoxication Phenotype

The intoxicated cohort had vital sign changes that would fit a pattern of potent α2-agonist exposure, with bradycardic heart rates (median = 54), and hypotensive blood pressures (median = 99/54). However, the most visible impact of medetomidine intoxication is the deep sedation that is unresponsive to naloxone [27]. Notably, while the cohort was small, nearly a quarter of our intoxication cohort required ICU admission, and almost as many required intubations due to the severity of their presentation. Although CNS and cardiovascular depression frequently occur during opioid intoxication and overdose, they usually resolve with naloxone administration. Our patients also experienced pronounced bradycardia and often hypotension, which correlates with the known mechanism of action of α2-agonists and supports other documented reports [28,29,30,31].

During the phase of intoxication, standard doses of naloxone had little to no effect on deeply sedated patients aside from restoring respiratory drive, likely related to concomitant fentanyl use. This is likely due to medetomidine’s sedative mechanism, which is independent of opioid receptors. This is analogous to xylazine intoxication, where patients may appear opioid-intoxicated but show variable improvement with naloxone [32,33]. Clinicians should be aware that a patient with suspected opioid overdose who remains unresponsive after adequate naloxone may have co-ingestion of a sedative. If their vital signs are markedly depressed, it is more likely that they are exposed to medetomidine, and sedation may be extended as compared to xylazine. Although naloxone does not directly target the α2 receptor, it is still recommended that providers administer it due to the likelihood of concomitant opioid intoxication, but to stop when respiratory drive is restored as not to precipitate severe opioid withdrawal.

If the patient remains heavily sedated with depressed hemodynamics, it is important to shift to supportive care: maintaining the airway (intubation if necessary) for oxygenation/ventilation and monitoring cardiovascular status [26,34]. There have been some case reports of dexmedetomidine in clinical settings causing extreme bradycardia and even cardiac arrest [35]. We did not encounter any cardiac arrests in our study period, but clinical risk and association should be investigated with further studies especially for patients with underlying heart disease.

Given that many medetomidine-positive patients also had xylazine exposure, it is worth comparing these two α2-agonists. The drug effects and profiles have only been studied in veterinary settings. Both cause sedation, bradycardia, hypotension, and analgesia via similar mechanisms [36,37]. Medetomidine is up to two hundred times more potent than xylazine with its binding affinity for central α2-receptors. The variable exposure and concentration of xylazine, compared to the markedly increased concentration of medetomidine metabolites in the urine of intoxicated individuals, support the contention that these severe findings are more associated with medetomidine exposure, and perhaps combined medetomidine-xylazine. Future studies should assess outcomes in individuals without cross-exposure to assess differences in intoxication outcomes.

### 3.4. Withdrawal Phenotype

In the withdrawal cohort, we observed a striking phenotype of adrenergic excess—characterized by hypertension, tachycardia, and agitation—with significantly elevated COWS scores (median = 23). This presentation is distinct from traditional opioid withdrawal, which typically lacks severe cardiovascular instability, and instead more closely resembles a sedative withdrawal. Medetomidine withdrawal syndrome shares pathophysiologic features with dexmedetomidine discontinuation and presents substantial challenges for ED and inpatient management. For example, medetomidine withdrawal was characterized by rapid, severe progression. We saw a precipitous shift in hemodynamics as demonstrated above with median pressures nearly 190/110 mm Hg, and even exceeding systolic pressures above 230 mg Hg. This sympathetic activation leads to hypertensive encephalopathy accompanied by refractory nausea/vomiting, tremors, and agony. Management of this unstable withdrawal syndrome was challenging and frequently required higher levels of care and a multimodal pharmacologic regimen [16]. Our findings align with recent alerts from the CDC and peer-reviewed reports documenting severe, prolonged, and treatment-refractory withdrawal in the context of α2-agonist co-exposure.

As mentioned above, xylazine was frequently detected in our cohort but did not appear to be the dominant driver of withdrawal severity. Notably, median xylazine concentrations were statistically similar in the withdrawal cohort and the intoxication group—a pattern that argues against xylazine being the primary cause of adrenergic crisis. This finding now potentially evolves the previous findings that xylazine was responsible for worsening withdrawal syndromes observed in some fentanyl users [38]. Nevertheless, without an available diagnostic test, the possibility of differentiating xylazine from medetomidine exposure is challenging.

### 3.5. Clinical Outcomes of Withdrawal

Withdrawal from medetomidine exposure was not only clinically severe but also logistically challenging. Nearly two-thirds of patients required ICU admission, and 16% were intubated during hospitalization. These rates exceed previously reported rates for fentanyl and xylazine withdrawal [39], underscoring the need for intensive hemodynamic monitoring and sedation strategies. Adjunctive α2-agonists such as clonidine or dexmedetomidine were often required, although no standardized treatment pathway exists yet. These management complexities are further compounded by high rates of patient-directed discharge (PDD) (~36%), suggesting either inadequate symptom control, psychological distress, or patients pre-contemplative to recovery, who leave as soon as symptoms are initially treated. Notably, patients who feel their symptoms are inadequately treated are often less engaged in recovery discussions and are more likely to direct their own discharge [40]. Clinicians should be aware that some patients receiving medications for opioid use disorder (MOUD)—particularly those on methadone—may be especially vulnerable to destabilization in the presence of medetomidine adulteration, as 23.8% of our cohort screened positive for methadone.

### 3.6. Limitations

This study is limited by its retrospective design, single-region scope, and reliance on a convenience sampling of patients with a limited number of available urine specimens. The small size of the intoxication cohort limits comparative inference, and causality cannot be established between medetomidine exposure and clinical symptoms due to polysubstance confounding and the lack of a control group. There were no reliable temporal use data, and only two cases with paired testing, to assess metabolic timelines or the impact of genetic polymorphisms. Collection of urine samples was unregulated and was delayed in a number of cases, preventing more widespread testing and limiting the accuracy of the analysis. Rates of positive immunoassay results may be falsely elevated, such as due to clinical medication provision, such as benzodiazepines and fentanyl, though neither is commonly used in either hospital for treatment of this condition. Nonetheless, the consistency of observed phenotypic features across cases, paired with universal-frequency metabolite detection and a robust chart abstraction protocol, strengthens the internal validity of our findings.

### 3.7. Summative Thoughts and Future Directions

In summary, medetomidine is rapidly metabolized to inactive forms and currently requires sophisticated analytical techniques for urine detection. Our findings, that attention to both the hydroxylation and glucuronidation pathways of metabolism is important for the detection of medetomidine exposure in human urine, provide the first direct evidence of medetomidine’s metabolic pathways being active in human subjects with non-medical opioid use. Our findings open several avenues for further research. Prospective studies are needed to validate the medetomidine-exposure clinical phenotypes and to establish standardized scoring tools and treatment algorithms for withdrawal. Pharmacokinetic testing in humans, using temporal and, possibly dose–response analysis, could inform optimal toxicology testing strategies. The development of rapid point-of-care medetomidine detection assays, including metabolite panels, is essential to facilitate real-time diagnosis and triage. Finally, public health surveillance systems should prioritize routine tracking of α2-agonists and other adulterants in fentanyl-associated overdoses and withdrawal cases, as their clinical impact appears broader and more dangerous than currently appreciated.

## 4. Materials and Methods

### 4.1. Study Design and Setting

We conducted a pragmatic retrospective observational study at two urban hospitals in Philadelphia, Pennsylvania, examining cases from 1 September 2024 to 30 April 2025. One is an academic hospital, a level 1 trauma center, and a tertiary referral center. The other is a community hospital 2.5 miles from the main hospital. Both sites serve large urban populations and care for a high volume of patients with opioid use disorder (OUD). This period followed the first detection of medetomidine in the local illicit opioid supply (documented in May 2024) and coincided with intensified surveillance for unusual overdose, intoxication, and withdrawal presentations. The study protocol was developed in concert with and adhered to the STROBE (Strengthening the Reporting of Observational Studies in Epidemiology) guidelines for observational research.

### 4.2. Study Population

Patients were screened for inclusion based on clinical presentations suggestive of *medetomidine exposure* in the context of illicit opioid use. Two primary presentation phenotypes were considered novel and indicative: (1) intoxication with marked CNS depression unresponsive to naloxone, accompanied by bradycardia (heart rate < 60 bpm) and/or hypotension (systolic BP < 90 mmHg) out of proportion to expected opioid effects; and (2) withdrawal with severe autonomic hyperactivity (tachycardia > 120 bpm and hypertension > 160/100 mmHg) and refractory vomiting, sometimes progressing to confusion or encephalopathic mental status not explained by typical opioid withdrawal alone. Importantly, several cases progressed from initial intoxication to withdrawal during their care and are reflected in both groups.

In practice, addiction medicine providers (physicians and trained advanced practice clinicians, APC) in the ED and hospital wards flagged suspected cases in real time. Patients were included if they were ≥18 years old, admitted history of fentanyl use, or had positive immunoassay testing for norfentanyl, and met one of the above syndromic (intoxication or withdrawal) definitions. We excluded cases where an alternative primary cause of symptoms was identified (e.g., sepsis, primary cardiac event, benzodiazepine intoxication for somnolent patients, or alcohol/benzodiazepine withdrawal for sympathetically stimulated patients), as well as cases where the patient had received clinical dexmedetomidine (to avoid misidentification of medical vs. illicit source) prior to urine toxicology acquisition.

### 4.3. Electronic Health Record Data Collection and Abstraction

Prior to data collection, the authors agreed on a codebook of described variables that could be reliably obtained in the electronic health record (EPIC Systems, Madison, WI). Once the codebook was designed, the authors (PD, SB, MC, DG, TWS, MM, MS, LO, MB, NT) manually abstracted charts for codebook data, with the senior author (KL) collating data and adjudicating any conflicting data by discussion. For each included patient, the authors abstracted clinical data from the hospital visit, including de-identified demographic data, vital sign abnormalities, disposition status, and additional clinical data. This included (1) use of α2-agonist medications to treat suspected medetomidine withdrawal, (2) severe outcomes (including myocardial infarction, defined by high-sensitivity troponin T (hsTnT, Roche Elecsys, diagnostic threshold of >53 ng/L), encephalopathy, seizures, and (3) maximum recorded COWS score [41]. Additionally, immunoassay toxicology screening results were also included, which reveal exposure to fentanyl, amphetamines, benzodiazepines, opiates, barbiturates, cocaine, cannabinoids, and methadone. Following coding of data, the identifying information was removed, and each patient was given a coded number, correlating to toxicologic analysis.

### 4.4. Specimen Collection

Given that standard toxicology screens do not test for medetomidine or xylazine, unused biological samples were retrieved, where available, and obtained within 72 h of arrival, and prior to clinical dexmedetomidine exposure (58/103 cases), for liquid chromatography–tandem mass spectrometry (LC-MS/MS) testing following conclusion of patient care. Some samples were collected promptly during treatment, allowing for accurate and timely analysis of urine in states of intoxication or withdrawal. In some instances, patients suffering from intoxication were unable to provide urine samples until they were in withdrawal; these were noted separately as ‘delayed’. All samples were de-identified, coded, and maintained frozen at −20 °C until batch analysis.

### 4.5. Toxicology Background and Analysis

The choice of LC-MS/MS targets was informed by prior studies indicating that hydroxylation at the 3-position of the medetomidine molecule and direct glucuronidation are the primary metabolic pathways in mammals [42,43]. Racemic (R/S; dex/levo) medetomidine is quickly absorbed after administration, with peak plasma levels occurring in approximately 30 min. Elimination from plasma is rapid, with reported half-lives varying between 0.96 and 1.28 h in dog and rat models [23,24], and 2–3 h in hours in humans [6]. In rats, absorption of medetomidine following SQ administration is rapid, with peak plasma concentrations reached within 10 min. Peak levels in the brain are five times higher than those in the plasma and are reached in 15–20 min. Approximately 85% of the drug in plasma is protein-bound. The excretion of radiologically labeled medetomidine is mainly in the urine, with less than five percent unmetabolized [25,44].

We therefore developed a targeted LC-MS/MS method to detect medetomidine’s expected metabolites in human urine. The method was based on a revised Research-Use-Only (RUO) LC-MS/MS method for xylazine [45,46]. The revised method involved the use of an expanded multiple reaction monitoring (MRM) table to include the list of analytes shown in Table 6, including 3-hydroxymedetomidine (3-OH-M), a main metabolite of medetomidine. Chromatography testing utilized a Phenomenex Kinetix C18 column (100 A, 5 µm, 50 × 4.6 mm) at 40 °C. Time-variable mobile phases (A = H_2_O, 0.1% formic acid; B = MeOH, 0.1% formic acid) were used at a fixed flow rate of 0.5 mL/min. The internal standard used was norfentanyl-d5 (Cerilliant Corp., Round Rock, TX, USA) for all metabolites. Preliminary analysis confirmed a few cases of unmetabolized medetomidine in urine samples, and hence, this testing was excluded from the final analysis. See Section A.1 (Figure A1, Figure A2, Figure A3, Figure A4, Figure A5, Figure A6, Figure A7, Figure A8, Figure A9, Figure A10, Figure A11) and Appendix A for standard curves, example chromatographs and result distributions.

All samples and standards were secondarily subjected to pretreatment with glucuronidase (B-One, Kura Biotech, Atlanta, GA, USA) according to the manufacturer’s procedure [47]. Glucuronidase pretreatment was chosen instead of direct detection of glucuronide metabolites due to cost and reagent availability considerations. Standard curves for each analyte utilize concentrations of 100, 50, and 0 ng/mL. Concentrations of analytes were interpolated from this single standard curve for measurements falling both within and above the range of standards. As a pilot study meant to identify drug exposure, the results reported in this study were those obtained from single runs using singleton samples. Positive results (analytes 1–3) were those for which numerical results were > 1 ng/mL. For 3-OH-M, positive results were those for which numerical results were > 1 ng/mL, or where the peak signal to baseline ratio was > 6.

### 4.6. Data Analysis

Due to the observational nature and lack of a control group, formal hypothesis testing was limited, as our aim was primarily descriptive. Therefore, demographic analysis was limited to summarized median patient characteristics (means for groups of two) and clinical characteristics using descriptive statistics and assessment of interquartile range for ages, vital signs, and COWS scores. To assess differences in toxicological findings between groups, we used Fisher’s exact test to compare binary detection rates (presence or absence of each analyte) and the Mann–Whitney U test to compare non-normally distributed analyte concentrations. Statistical significance was defined as a two-tailed *p*-value < 0.05. These were performed using Microsoft Excel (Microsoft, Seattle, WA, USA) and R Version 4.4.3 software (R Core Team, Vienna, Austria). Intoxication and withdrawal presentations were compared separately, though some patients who presented with intoxication later developed withdrawal (see Figure 1).

Additionally, individuals who presented intoxicated but whose urine was collected during the withdrawal period were analyzed with the withdrawal subset. Analysis included both the frequency of substance detection and the quantifiable amount of the substance found in the urine LC-MS/MS testing sample.

## 5. Conclusions

Medetomidine has rapidly entered the illicit opioid scene as a high-potency adjunct to fentanyl, and this study elucidates the clinical consequences of its use in humans. In a Philadelphia cohort, we identified a characteristic pattern: medetomidine contributes to intoxication and overdose marked by bradycardia and sometimes hypotension with profound sedation unresponsive to naloxone, and it precipitates a severe withdrawal syndrome of autonomic hyperactivity when its effects wear off. These presentations can be differentiated from classic opioid or sedative intoxication by their unique clinical features and require tailored management strategies (emphasizing supportive care for overdose and aggressive α2-agonist therapy for withdrawal). Using advanced LC-MS/MS analysis, we confirmed medetomidine exposure via its urinary metabolites, marking the first biochemical documentation of medetomidine exposure in patients suffering from withdrawal, and mirroring metabolic pathways known from animal studies.

Our findings carry important public health implications. As medetomidine proliferates in the drug supply, frontline medical providers must adapt to evolving toxidrome and withdrawal scenarios, and laboratories must plan to update toxicology screens accordingly. There is an urgent need for increased surveillance, education, and perhaps new treatment protocols to address these emerging sedative intoxication and withdrawal syndromes. Ongoing research should monitor the spread of medetomidine, evaluate long-term outcomes in those exposed, and investigate optimal therapies (for example, the safe use of dexmedetomidine or other agents to manage withdrawal). Finally, our study underscores a broader point: the illicit drug supply continues to shift in unpredictable ways, and a robust interdisciplinary approach—combining clinical observation, laboratory detection, and pharmacologic insights—is key to identifying and combating novel adulterants like medetomidine. By sharing these early insights, we hope to improve recognition and care of medetomidine-exposed patients and inform broader efforts to mitigate harm in the fentanyl era.

## Figures and Tables

**Figure 1 ijms-26-06715-f001:**
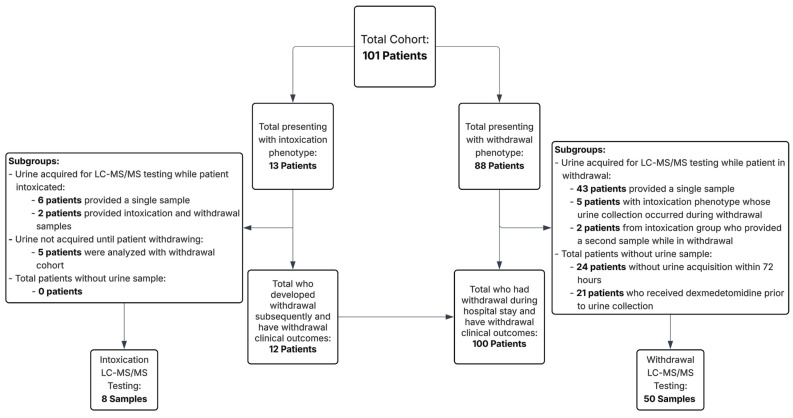
Patient and toxicology sample cohort by presentation phenotype and urine acquisition for liquid phase chromatography–tandem mass spectrometry (LC-MS/MS) testing.

**Table 1 ijms-26-06715-t001:** Demographics and outcomes—demographics, usage patterns, vital signs, disposition, and substance co-detection by presentation type.

Characteristic	Intoxication Phenotype (*n* = 11)	Withdrawal Phenotype (*n* = 88)	Paired Samples (*n* = 2)
Age, median (mean for paired) years (IQR or range)	39 (36–53)	39 (35–44)	61.5 (59.0–64.0)
Male sex, *n* (%)	4 (36.4%)	71 (80.7%)	1 (50%)
Female sex, *n* (%)	7 (63.6%)	17 (19.3%)	1 (50%)
Reported Daily Use Volume, bags/day (IQR)	15 (5–25)	15 (8–29)	4.5 (1–8)
Route: Injection, *n* (%)	11 (85%)	47 (52%)	1 (50%)
Route: Insufflation, *n* (%)	3 (23%)	32 (36%)	2 (100%)
Route: Smoke, *n* (%)	2 (15%)	8 (9%)	0 (0%)
Route: Not recorded, *n* (%)	0 (0%)	7 (8%)	0 (0%)
Median Max/Min Heart Rate, bpm (IQR)	54 (50–58)	132 (112–147)	*
Median Max/Min Systolic BP, mmHg (IQR)	99 (79–102)	190 (168–210)	**
Median Max/Min Diastolic BP, mmHg (IQR)	54 (38–58)	110 (99–127)	***
ED Disposition: Discharged, *n* (%)	1 (8%)	1 (1%)	0 (0%)
ED Disposition: Admitted, *n* (%)	12 (92%)	87 (98.9%)	2 (100%)
Hospital Disposition: Discharged, *n* (%)	8 (62%)	24 (27.2%)	1 (50%)
Hospital Disposition: PDD, *n* (%)	4 (31%)	32 (36.4%)	1 (50%)
Hospital Disposition: Rehab/Detox, *n* (%)	1 (8%)	21 (23.9%)	0 (0%)
Hospital Disposition: Law Enforcement, *n* (%)	0 (0%)	13 (14.8%)	0 (0%)
Hospital LOS, mean days (IQR)	13 (5–15)	7 (3 –11)	10 (7–13)
ICU admission, *n* (%)	3 (23.1%)	56 (63.6%)	1 (50%)
Required intubation, *n* (%)	2 (15.4%)	12 (13.6%)	1 (50%)
Xylazine-positive, *n* (%)	7 (53.7%)	31 (62%)	1 (50%)
Benzodiazepine-positive, *n* (%)	4 (30.8%)	36 (36%)	2 (100%)

* Paired Cohort Median Min/Max Heart Rate, bpm (IQR): 53 (48–58)|162 (120–204). ** Paired Median Max/Min Systolic BP, mmHg (IQR): 112 (93–132)|208 (194–221). *** Paired Median Max/Min Diastolic BP, mmHg (IQR): 58 (54–61)|120 (103–137).

**Table 2 ijms-26-06715-t002:** Standard urine immunoassay toxicology summary demonstrating high burden of exposure to multiple substances, including cocaine, benzodiazepines, amphetamines, and methadone.

Urine Drug Test	Positive, *n* (%)
Fentanyl	100 (100.0%)
Cocaine	60 (60%)
Opiates	40 (40%)
Benzodiazepine	37 (37%)
Amphetamine	35 (35%)
Methadone	24 (24%)
Cannabinoid	14 (14%)
Barbiturates	8 (8%)

**Table 3 ijms-26-06715-t003:** LC-MS/MS toxicology result comparison by clinical phenotype demonstrating universal fentanyl and medetomidine exposure (measured via Norfentanyl and 3-OH-M post-glucuronidase testing) and variable xylazine exposure.

Analyte	Intoxication (*n* = 8)	Withdrawal (*n* = 50)	*p*-Value (Fisher)	*p*-Value (Mann–Whitney)
Fentanyl, *n* (%)	8 (100%)	34 (68%)	0.092	
Median Fentanyl, ng/mL (IQR)	1107 (406–1858)	39 (0–255)		0.001
Norfentanyl, *n* (%)	8 (100%)	50 (100%)	1.000	
Median Norfentanyl, ng/mL (IQR)	9666 (2302–18,929)	741 (312–1769)		0.001
3-OH-M, *n* (%)	8 (100%)	34 (68%)	0.092	
Median 3-OH-M, ng/mL (IQR)	561 (233–753)	13 (0–80)		<0.001
3-OH-M post-Glucuronidase, *n* (%)	8 (100%)	50 (100%)	1.000	
Median 3-OH-M post-Glucuronidase, ng/mL (IQR)	8423 (375–1311)	34 (10–166)		<0.001
Xylazine, *n* (%)	4 (50%)	25 (50%)	1.000	
Median Xylazine, ng/mL (IQR)	4 (0–11.2)	3 (0–43)		0.691
Xylazine post-Glucuronidase, *n* (%)	8 (100%)	32 (64%)	0.048	
Median Xylazine post-Glucuronidase, ng/mL (IQR)	24 (11–73)	11 (0–75)		0.272

**Table 4 ijms-26-06715-t004:** LC-MS/MS toxicology result comparison for paired intoxication/withdrawal samples demonstrating rapid elimination of fentanyl, norfentanyl, and medetomidine.

	Intoxication vs. Withdrawal	Time Between Samples (h)	Fentanyl (ng/mL)	Norfentanyl (ng/mL)	3-OH-M (ng/mL)	3-OH-M Post-Glucuronidase (ng/mL)	Xylazine (ng/mL)	Xylazine Post-Glucuronidase (ng/mL)
Patient 1								
	Intoxication	--	197	1131	144	154	7	8.0
	Withdrawal	48.0	0	330	7	10	0	0.0
Patient 2								
	Intoxication	--	1446	20,029	1070	1241	0	69.0
	Withdrawal	13.1	425	3514	106	502	0	21.0

**Table 5 ijms-26-06715-t005:** Withdrawal clinical outcome summary. A tally of clinical treatment and outcomes for those suffering from withdrawal symptoms from the cohort. This includes patients with primary intoxication phenotype who later developed withdrawal symptoms.

Outcome	All Patients (*n* = 100)
Dexmedetomidine Infusion	63 (63%)
Clonidine PO use	89 (89%)
Clonidine Transdermal Patch use	78 (78%)
NSTEMI/hsTnT Elevation > 53 ng/L	20 (20%)
Encephalopathy as diagnosis or PRES	31 (31%)
Seizure During Visit	5 (5%)
Highest COWS Score	23 (19–28)

**Table 6 ijms-26-06715-t006:** MRM table Q1/Q3 represents the positive-ion–ion pairs used in analyte detection.

ID	Analyte	Q1 (*m*/*z*)	Q3 (*m*/*z*)
1	Fentanyl	337.1	117.2
2	Norfentanyl	233.2	84.2
3	Xylazine	221.0	90.1
4	3-OH-M	217.1	68.2
5	Norfentanyl-d5	238.2	84.2

## Data Availability

Data are available online, adjacent to this publication.

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
