# Peer review of "Biochemical Identification and Clinical Description of Medetomidine Exposure in People Who Use Fentanyl in Philadelphia, PA"

_ijms, 2025, doi:10.3390/ijms26146715_

Round 1
Reviewer 1 Report
Comments and Suggestions for Authors
This article contains important and useful information about intoxication caused by medetomidine in street fentanyl samples. The reviewer recommends to accept it after minor correction. Some opinions are listed below.
- Figure 1. This flow chart is difficult to understand. At least, it is difficult to understand difference between sample numbers and patient numbers.
- The authors set 1 ng/mL as the clinical cut-off concentration. On the other hand, the authors used two-point calibration curves not including 1 ng/mL. If the authors want to set any cut-off concentration, they should contain the corresponding calibration point.
- Medetomidine may give N-glucuronides as one of the metabolites. Such metabolites may be detected as medetomidine after enzymatic hydrolysis. Why didn’t the authors target it?
Please add discussion about medetomidine concentration in street fentanyl samples. Considering higher pharmacological activity of medetomidine than xylazine, if the medetomidine concentration is similar to that of xylazine, it may lead severe results.
Author Response
We are so grateful for your review! Please see below for a point by point response to your comments:
- Figure 1. This flow chart is difficult to understand. At least, it is difficult to understand difference between sample numbers and patient numbers. This is fair and I have reformatted it to be clearer when it's a patient or sample
- The authors set 1 ng/mL as the clinical cut-off concentration. On the other hand, the authors used two-point calibration curves not including 1 ng/mL. If the authors want to set any cut-off concentration, they should contain the corresponding calibration point. I think this is a misunderstanding, 1 ng/mL is included in the calibration curves, this may be an artifact of how the curves printed. I'm sorry for the confusion.
- Medetomidine may give N-glucuronides as one of the metabolites. Such metabolites may be detected as medetomidine after enzymatic hydrolysis. Why didn’t the authors target it? Great question, the reagents required to detect the glucuronides were not available and outside the budget of the project. I have added statements (in track changes) to the methods and discussion to address this. Thank you!
Please add discussion about medetomidine concentration in street fentanyl samples. Considering higher pharmacological activity of medetomidine than xylazine, if the medetomidine concentration is similar to that of xylazine, it may lead severe results. I would prefer to have more data. I think we discuss how it is the potency of medetomidine rather than xylazine that is contributing, but I don't have enough granular drug checking data to feel confident in the above (just human sample data).
Reviewer 2 Report
Comments and Suggestions for Authors
-Line 52: PA Groundhogs, use full form
-Line 82-95: Authors have described about the results, but at the end of introduction, aim and objective of the study should be mention instead. write down aim here.
-In Method: what standard and column were used for LC-MS analysis.
-Whats the rationale to apply Fisher test over Mann-Whitney?
Author Response
Thank you again for your attention and willingness to review our paper. We appreciate your suggestions, which improve the strength of the piece. Track changes were used, see below for a point by point response.
-Line 52: PA Groundhogs, use full form The name of the organization is actually PAGroundhogs, I have removed the space, so the name is correct. Extending it to "Pennsylvania Groundhogs" would be misnaming the organization.
-Line 82-95: Authors have described about the results, but at the end of introduction, aim and objective of the study should be mention instead. write down aim here. I agree it was not clearly stated, I have rewritten the last paragraph of the introduction.
-In Method: what standard and column were used for LC-MS analysis. Thank you for helping us improve the clarity and content. While the standards were described, the column was not, these were moved next to each other in the methods.
-Whats the rationale to apply Fisher test over Mann-Whitney? This is addressed in lines 198/199, the Fisher test was for binary values (such as whether drug was present or not), MW was for (non-normally distributed) concentrations. If there is somewhere else this would be helpful to state, we appreciate an opportunity to improve clarity.
Round 2
Reviewer 2 Report
Comments and Suggestions for Authors
Line 159: It is stated "Racemic" , does it + or - racemic one?
Line 204: Still country name is missing
Line 358: COWS was used in method section before, use of full form should be written only once, later should use abbreviation.
Author Response
Comment 1: Added.
Comment 2: Great catch, added.
Comment 3: I even found a 3rd, removed both instances following initial definition.
Thank you for strengthening our paper!